# Are Renal Cell Carcinoma with Fibromyomatous Stroma (RCC-FMS) and Thyroid-like Follicular Carcinoma of the Kidney (TLFCK) Really Independent Variants?

**DOI:** 10.3390/diagnostics13010086

**Published:** 2022-12-28

**Authors:** Stefano Stanca, Laura Boldrini, Paola Anna Erba, Pinuccia Faviana

**Affiliations:** 1Department of Surgical, Medical, Molecular Pathology and Critical Area, University of Pisa, Via Savi 10, 56126 Pisa, Italy; 2Department of Translational Research and New Technologies in Medicine and Surgery, University of Pisa, Via Savi 10, 56126 Pisa, Italy

**Keywords:** emerging entity, kidney, new entity, renal cell carcinoma

## Abstract

Background: Renal cell carcinoma with fibromyomatous stroma (RCC-FMS) is a recent provisional entity already recognised in the 2016 WHO Classification of Cancer of the Urinary Tract and Male Genital Organs 4th Edition as renal cell carcinoma with (angio)leiomyomatous stroma, histologically defined as a tumour characterised by clear cells intertwined in a conspicuous vascular stroma. In the casuistry taken into consideration, another proposed variant, thyroid-like follicular carcinoma of the kidney (TLFCK), endowed with a morphology mimicking thyroid parenchyma, was examined. The aim of this work was to parse the theoretical system, experimental data and diagnostic impact of these new entities proposed in the field of renal neoplasms. Materials and Methods: An analysis of 120 cases of kidney tumours from the Department of Surgical, Medical, Molecular and Critical Area at the University of Pisa was run. Subsequently, all samples were reassessed by two pathologists with expertise in uropathology, whose revaluation provided a histomorphological study combined with subsequent and coherent immunohistochemical analyses of CK7, CD10, CAIX, CK34betaE12, CD117, vimentin, TTF-1 and thyroglobulin. These analyses were performed using the Ventana Benchmark Automated Staining System (Ventana Medical Systems, Tucson, AZ, USA) and Ventana reagents. Results: On the one hand, the data, thus brought to light, did not show an immunohistochemical profile consistent with that proposed for RCC-FMS. However, it should be emphasised that the morphological background also unearthed a poor specificity for RCC-FMS. This was specifically due to a stromal component which was, in any case, evident, although characterised by a wide range of presentation, in clear cell renal cell carcinoma (ccRCC). This latter is, indeed, the reference background for this theorised variant. On the other hand, a thyroid-like pattern was highlighted in 11 cases, more specifically in 10 ccRCCs and in one oncocytoma, presenting itself as a type of neoplastic appearance rather than as the peculiar morphological pattern of a standalone cancer. Conclusions: In the light of these results, RCC-FMS and TLFCK appear to be more appropriately variants of already categorised neoplastic entities rather than new independent neoplasias.

## 1. Introduction

Renal cell carcinoma with fibromyomatous stroma (RCC-FMS) is a recent provisional entity already recognised in the 2016 WHO Classification of Cancer of the Urinary Tract and Male Genital Organs 4th Edition as renal cell carcinoma with (angio)leiomyomatous stroma [1]. It is histologically defined as a tumour characterised by clear cells intertwined by a prominent vascular and smooth muscle stroma [1,2].

The leiomyomatous stroma is, however, not an unusual finding in clear cell RCC (ccRCC) [2,3,4,5]. 

In this postulated variant, neoplastic cells with low grade nuclei, often without nucleoli (ISUP grade 1–2) [3], marked by a pale and sometimes eosinophilic cytoplasm and surrounded by a definite membrane are structured according to several cytoarchitectural patterns from alveolar to acinar, in an interweaving of a sinusoidal and pleomorphic stroma [4]. This neoplastic scenario is deprived of expansive growth, necrosis, vascular invasion or sarcomatoid alteration [2,6]. Furthermore, RCC and particularly clear cell renal cell carcinoma (ccRCC) can even show multiple fluid-filled cysts with a network of fibromuscular and hypercellular stroma, resembling the thyroid parenchyma [7,8,9], featured by atrophic renal tubules [9], dispersed glomeruli [10,11] and variously represented fibrotic stroma [11]. 

Stromal elements vary in cellularity, but, at the same time, lack areas of cytological atypia, necrosis or increased mitoses [11,12]. Renal tubules can be either incorporated within the stroma, thus forming a glandular pattern mimicking nephrogenic adenoma, or exhibiting thyroid characteristics with accumulation of thickened eosinophilic material [11,13]. 

This is the background of thyroid-like follicular carcinoma of the kidney (TLFCK) [11].

In our study, we therefore considered a group of renal tumours, focusing both on those endowed with clear cell cytology against the backdrop of leiomyomatous stroma and with thyroid-like morphology. 

The immunophenotypic stains examined in an attempt to select these new variants were, as highlighted and discussed, not diriment. Given these premises, can RCC-FMS and TLFCK really be considered two new entities or, more appropriately, a different histological presentation pattern, seeing that no prognostic and survival impact has been mentioned in literature?

## 2. Materials and Methods

The aim of this study was to structure a critical synopsis of the theoretical framework in which RCC-FMS and TLFCK were born as alleged new entities.

The study is a combination of an experimental approach and a literature review. 

The first involved the morphological and immunohistochemical analysis of our casuistry, the second a deep examination of the state of the art. This latter included not only a morphological and immunohistochemical literature data assessment, but also a molecular data evaluation, not present in the experimental work. 

Conclusions will be drawn consistently with the integration of both approaches. 

This work embraced the analysis of 120 cases of kidney tumours from the Department of Surgical, Medical, Molecular and Critical Areas of the University of Pisa, since the end of 2016 to the beginning of 2022.

In particular, 

-90 ccRCCs, of which one included a rabdoid component and two with sarcomatoid features,-Three clear cell papillary renal cell carcinomas (ccpRCC),-Eight papillary carcinomas type 1 (pRCC Type 1),-Nine papillary carcinomas type 2 (pRCC Type 2),-Eight oncocytomas,-Two chromophobe carcinomas (chRCC)

were reassessed firstly from a morphological point of view. 

Those morphologically suggestive of RCC-FMS and/or for TLFCK were therefore studied through immunohistochemistry, in detail, by CK7 [2,13,14,15], CD10 [2,13,15], CAIX [2,13,15], CK34betaE12 (as a marker of high molecular weight cytokeratin (HMWC)) [2,13,15], CD117 [13], vimentin [2,13,15], TTF-1 [13,15,16] and thyroglobulin [13,15,16], using the Ventana Benchmark Automated Staining System (Ventana Medical Systems, Tucson, AZ, USA) and Ventana reagents. All samples were re-evaluated by two pathologists with expertise in uropathology. 

The study was conducted in accordance with the Declaration of Helsinki, and approved by the Ethical Committee of the Area Vasta Nord-Ovest (CEAVNO) (protocol code 9989, 20 February 2019).

This methodology is, in our opinion, original, since it represents the unique case in peer reviewed open literature of a systematic analysis of previous diagnostic cases to detect the criteria of RCC-FMS and TLFCK diagnosis.

The reference sample of 120 cases, even though limited, was nevertheless greater than those in the literature taken into consideration and acts as the empirical counterpart of the critical analysis here conducted on works of literature. 

## 3. Results

The data obtained did not highlight on any occasion an immunohistochemical profile coherent with that proposed for RCC-FMS (diffuse CK7+, diffuse membranous or cup-shaped CAIX+, CD10+, CK34betaE12+) [13]. 

With regard to this, it has to be emphasised that even the morphological background did, in these cases, reveal a poor specificity for RCC-FMS, considering that the stromal component is represented in ccRCC anyway. 

On the contrary, TLFCK was evidenced in 11 cases out of 120 (9.16%), in particular in 10 ccRCCs and in one oncocytoma with an age range between 47 and 79 yrs. In these histotypes, thyroid-like morphology was about 70–80%. 

To be precise, here the cases that perfectly summarise the achievements are reported: a case previously diagnosed as ccRCC, marked by a considerable stromal component, revealed the following significative pattern: patchy CK7+, box-shaped CAIX+, vimentin+, CK34betaE12−, CD10−/+. Moreover, it was characterised by cystic areas, in correspondence of which, CK7 was extremely positive, against other neoplastic areas of complete negativity (Figure 1);a case formerly classified as ccRCC endowed with an important neoplastic stroma and thyroid-like areas, displayed: patchy CK7+, predominantly cup-shaped CAIX+, CK34betaE12CK34betaE12−, CD10+++, TTF-1-, thyroglobulin- (Figure 2);a case diagnosed as ccRCC showed: CK7−, box-shaped CAIX+, CK34betaE12CK34betaE12−, CD10+++, TTF-1-, thyroglobulin- (Figure 3);a case assessed as oncocytoma, was featured by CK7−, CD117+, CK34betaE12CK34betaE12−, vimentin−/+, TTF-1-, thyroglobulin- (Figure 4).

## 4. Discussion

The aim of this work was to investigate the theoretical system, experimental data and diagnostic impact of two novel entities, RCC-FMS and TLFCK, proposed within the framework of renal neoplasms, although not counted, and according to us rightly, in the classification provided by the 2022 WHO. 

In order to achieve this milestone, it was essential to clarify the biological assumptions and clinical consequences implying the introduction of new neoplastic entities. We combined the analysis of our casuistry with literature results on the matter. 

A synopsis of the status question is and an attempt to understand if and how clinical practice could change in the light of the new supposed diagnostic models were thus offered through this approach. 

New variants should, in fact, satisfy, firstly, the specific morphological and immunohistochemical criteria able, on the one hand, to define a precise histotype and molecular profile of neoplasia, and on the other, to propose an identikit of markers with potential therapeutic and prognostic repercussions. 

We specifically analysed these new entities whose existence, independent of other diagnostic categories already put forward as emerging and provisional in the 2016 WHO, was primarily postulated in the ccRCC background for RCC-FMS and mainly in the ccRCC and oncocytoma framework for TLFCK [13].

On the basis of these introductory notes, the most important question to be answered is whether RCC-FMS and TLFCK are new carcinomas in the kidney neoplastic scenario or if, for them, another type of categorisation would be more appropriate.

In this respect, a retrospective analysis to investigate the histological, cellular and immunophenotypical surroundings of these postulated neoplasias was performed. 

In the first place, in our casuistry, as follows from the obtained results, neither histological nor immunohistochemical characteristics evidenced a diriment pattern for RCC-FMS. 

Its architecture embraces a wide range of morphologies from ramified tubules, solid nests to papillary and cystic structures, reflecting mainly ccRCC histological configuration [14] but also a ccpRCC configuration [17], intertwined by a thick stromal network [2,3,14]. 

However, by virtue of the multifaceted and blurred nature of ccRCC, both from a cellular and stromal point of view, neither the cellular architecture nor the fibroleiomyomatous component are sufficient for RCC-FMS diagnosis [18].

The stromal component in ccRCC and in ccpRCC covers, in fact, a wide range of expression. 

On this basis, can a more representative stroma in a cancer background already classified, such as ccRCC and ccpRCC, endowed with a defined prognosis, configure another distinct entity, without, in our view, sufficient evidence of specific biological markers and a different clinical behaviour? 

Consequently, immunohistochemistry was necessary in the attempt to trace its biological definition.

RCC-FMS immunohistochemical profile requires positivity for CK7, CAIX, CD10 and a differential diagnosis with ccRCC and ccpRCC. In addition, CD10 and high molecular weight cytokeratin (HMWC) positivity have also been considered as an RCC-FMS characteristic [2]. 

In the first instance, positivity for CK7 is extremely variable in the ccRCC behavioural spectrum [19]. In particular, a prevalent negativity up to a focal positivity has been reported, not only in high grade ccRCC with eosinophilic cellularity [20], but also in low grade ccRCC [21] and, although not common, also a widespread positivity [19]. As confirmation of the protean expression of CK7 in ccRCC, it is worth mentioning its diffuse positivity in the cystic component of ccRCC [22]. 

This variability also emerged in our wide casuistry: in the two mentioned ccRCC cases with a stromal structure more suggestive of RCC-FMS, CK7 actually presented negativity contextually to positive areas, particularly in the cystic spots, in accordance with literature data. As we shall see shortly, despite their evocative histological structure, immunohistochemical results, proposed as essential for RCC-FMS diagnosis, will not be diriment in this sense. 

CK7’s contextual positivity in relation to negativity or focal positivity for CD10 [23] has always represented a typical immunophenotypical feature of ccpRCC [21,24,25,26,27,28]. This is a variant whose positivity for CK7, with an associated negativity for CD10, AMACR and RCC has been widely documented [22], together with its indolent clinical behaviour [29].

Furthermore, it is necessary to remember, in secundis, that CD10 itself, although predominantly negative in ccpRCC [30], can also be diffusely [31] or focally positive [28] and can show, in these cases, a reverse cup-shaped staining [31].

Thirdly, in ccRCC, the staining for CAIX is positive, predominantly diffusely, but also focally [31], with a prevalent, although not exclusive, box-shaped pattern; on the contrary, in ccpRCC it is positive with a predominant cup-shaped expression [32]. Lastly, HMWC has been described in the literature as up to 100% positive in papillary carcinomas, more rarely in clear cell carcinoma [33], and diffusely in RCC-FMS [34]. CD10 is positive in ccRCC [35] and in RCC-FMS, but negative or focally positive in ccpRCC [36], an irregularity in staining that has been brought to light in the literature as well as in our results, where CD10 evidenced a diffuse and, at the same time, patchy positivity in a ccRCC background.

According to us, in agreement with our results combined with the previous data, RCC-FMS, characterised by a clinical course indistinguishable from ccRCC [2] and a favourable prognosis [2,13], would not satisfy sufficient diagnostic criteria to be viewed as an autonomous diagnostic entity, as previously sustained, on the grounds of it not having peculiar morphological and immunophenotypical traits, shared mainly with ccRCC [37,38] and ccpRCC [39].

In support of this position, in fact, there have been documented cases of RCC marked, together with RCC-FMS, by a contextual expression of ccRCC and ccpRCC morphological features with the respective immunophenotype [40]. Not only this, but there have also been described ccRCCs with a hybrid combination of borderline morphological patterns, namely gland/nest-like and papillary and immunomarkers such as negative or focal/>50% of neoplastic cells with CK7 positivity, typical in ccpRCC, and CD10 and CAIX positivity, but peculiar in ccRCC [19].

In our casuistry, in the examined ccRCC cases with an RCC-FMS-like histotype, not only CK7, as already discussed, but also CAIX (weak and variable from box- to cup-shaped), CK34betaE12 (negative) and CD10 (patchy positive) were incongruent with RCC-FMS diagnosis, on the one hand, and confirmatory of the non-diriment histology, on the other, of the multiform histochemical response of ccRCC.

Though molecular analysis was not the object of the experimental investigation, representing a limitation of this work, the RCC-FMS genetic pattern would be defined by TSC-mTOR and ELOC/TCEB1 mutations with VHL wild-type without 3p deletion [13].

However, following a literature review on the question, it was found that, in the face of the TCEB1-mutant RCC with a thick fibrous component [41], non-TCEB1 mutant ccRCCs with the same morphological habitus have also been documented [42]. In addition, ccRCC with wild-type VHL and TCEB1 mutations has been reported as an aggressive metastatic entity characterised by a sarcomatoid differentiation [43] thus manifesting an opposed behaviour to RCC-FMS, with which it shares molecular mutations, but not the prognosis. Furthermore, TSC-mTOR mutations have been associated with RCCs with different histological architectures and cytological features such as papillary [44,45], oncocytic [46,47,48,49] and chromophobe [44,45]. These mutations result in an mTOR-pathway activation [50], that is, at the same time, widely reported in ccRCC [51].

Hence, an attentive review of molecular analyses on RCCs in literature, due to its seminal and essential role in profiling new neoplasias, did not highlight a clear and incontrovertible correlation between specific mutations and a distinctive histologic and immunohistochemical pattern.

For the above reasons, even molecular analysis does not seem to have proved decisive in the identification of RCC-FMS in the background of ccRCC.

To sum up the state of art addressed above,

-RCC-FMS is histological contiguous with ccRCC and ccpRCC,-It has a diffuse positivity for CK7, by analogy with ccpRCC, although also with high grade ccRCC,-Its CD10 positivity is shared mainly with ccRCC, but also with a reduced amount of ccpRCC cases,-RCC-FMS is widely box-shaped, but also has cup-shaped positivity for CAIX, similar to ccRCC and ccpRCC,-Finally, an indolent clinical course classifying RCC-FMS as a new histological and clinical entity is difficult to sustain and not particularly helpful in clinical practice.

To conclude, our position is that it would be more suitable for RCC-FMS to be named ccRCC-FMS, standing for clear cell renal cell carcinoma with fibromyomatous stroma as a variant of an already classified, although histological polyhedric, neoplastic entity.

On the other hand, the further entity addressed, TLFCK, is devoid of a peculiar immunohistochemical profile [52].

In the cases investigated, several neoplasms, previously diagnosed as ccRCC, showed a thyroid-like aspect, characterised by an eosinophilic content cystic-follicular architecture, with negativity for TTF-1 and thyroglobulin and without, as already mentioned, a specific immunohistochemical pattern. By the same token, our study highlighted the wide age range of expression, even though literature data, in particular from Chinese case reports, noted a prevalent young age of incidence.

Given our results, the exiguity of TLFCK cases documented since 2004, less than 40 [13,18], perplexed us. Indeed, our results evidenced a frequency of 9.16% out of 120 RCC cases from 2016 to 2022.

In this regard, it is, therefore, our position that as a type of neoplastic presentation rather than an independent neoplastic entity, its real incidence has been underestimated.

From this point of view, our hypothesis of a TLFCK lack of biological independence is confirmed by the controversial immunohistochemical results, revealing, for example, positivity [53] and negativity for CK7 in addition to a frequent negativity for CD10 and RCC [54], typical markers of ccpRCC. This is, according to us, due to the overlapping variability of the histological framework, from tubulocystic to solid and papillary, in which a thyroid-like lesion can be detected: from ccRCC to ccpRCC, pRCC, oncocytoma and the histological combination of these last two entities, the oncocitic variant of papillary renal cell carcinoma [55,56,57] with consequent different and specific immunohistochemical responses.

In the face of its benign behaviour, metastases have also been described in a case of diagnosis of TLFCK with CK7, CK19, vimentin positivity and RCC and CD10 negativity [15].

TLFCK is a histologic pattern, for which the absence of a molecular signature has been documented because its morphological habitus is common to different neoplastic entities.

At the same time, TLFCK is hardly distinguishable from atrophic kidney-like lesion (AKLL), a neoplastic category endowed with larger follicular areas, but bereft of a specific immunohistochemical pattern. AKLL has been described in a reduced number of studies, highlighting a negative staining for PAX-8, that, similar to podocytes, would imply the origin of cysts in a glomerular dilatation and atrophy.

Regarding this matter, a case of TLFCK in association with autosomal dominant polycystic kidney disease (ADPKD) in a 34-year-old subject has also been reported. It was characterised by a cystic architecture, nevertheless typical of every renal cancer arising on a polycystic background, with colloid/mucoid-like endoluminal material and a Fuhrman grade 2 and 3 [58]. However, by comparing this mass–lesion to a normal kidney affected by glomerulocystic disease (GCKD), the morphological and immunophenotypical presentations have been identical [59] even while considering that thyroidisation also occurs in inflammatory diseases such as pyelonephritis [60].

Accordingly, and as also confirmed by our study, thyroid habitus seems to be a histologic eventuality, a bridge between several pathological entities, neoplastic and not, whose many-sidedness is associated with pleomorphic biological behaviours and overlapping molecular responses.

In the light of these data, further studies will serve in the future to clarify the eventual independent nature of these neoplastic variants proposed as emerging provisional entities in the 2016 WHO, but not accepted in 2022.

## 5. Conclusions

As specified at the beginning, this article started from the combination of an experimental design and a literature review.

As we have seen, the peculiarity of this study, which defines its difference with the other methodological approaches in literature, was the building of a vision of the whole.

Indeed, its fundamental characteristic was to compare the several diagnostic requirements for RCC-FMS and TLFCK in literature, investigating and, consequently, calling into question their absolute legitimacy.

At present, the redundant puzzle of the morphological, immunophenotypical and molecular features of RCC-FMS and TLFCK appears to noticeably support noticeably the idea that they are mere ways of neoplastic presentation in the ccRCC and in the ccRCC/oncocytoma/papillary carcinoma spectrum respectively, rather than new neoplastic subjects endowed with an independent nature.

By the same token, by virtue of the absence of a different treatment, their diagnostic helpfulness has yet to be demonstrated.

To conclude, the polyhedric picture emerging from the literature and confirmed by the results reached in this work underlines the lack of criteria sufficient to define RCC-FMS and TLFCK as diagnostic novelties.

## Figures and Tables

**Figure 1 diagnostics-13-00086-f001:**
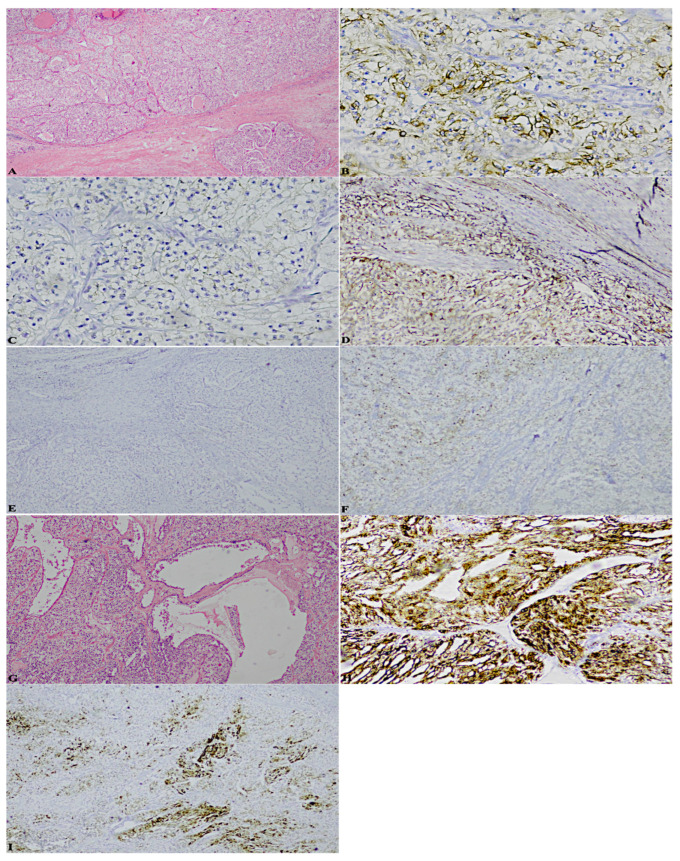
(**A**). Hematoxylin-eosin neoplastic stroma (2×) (**B**). Patchy CK7+ (10×) (**C**). Box-shaped CAIX+ (10×) (**D**). Vimentin+ (10×) (**E**). CK34betaE12− (4×) (**F**). CD10−/+ (4×) (**G**). Hematoxylin-eosin ccRCC cystic areas (10×) (**H**). ccRCC cystic areas CK7 +++ (10×) (**I**). ccRCC cystic areas CK7−/+ (4×).

**Figure 2 diagnostics-13-00086-f002:**
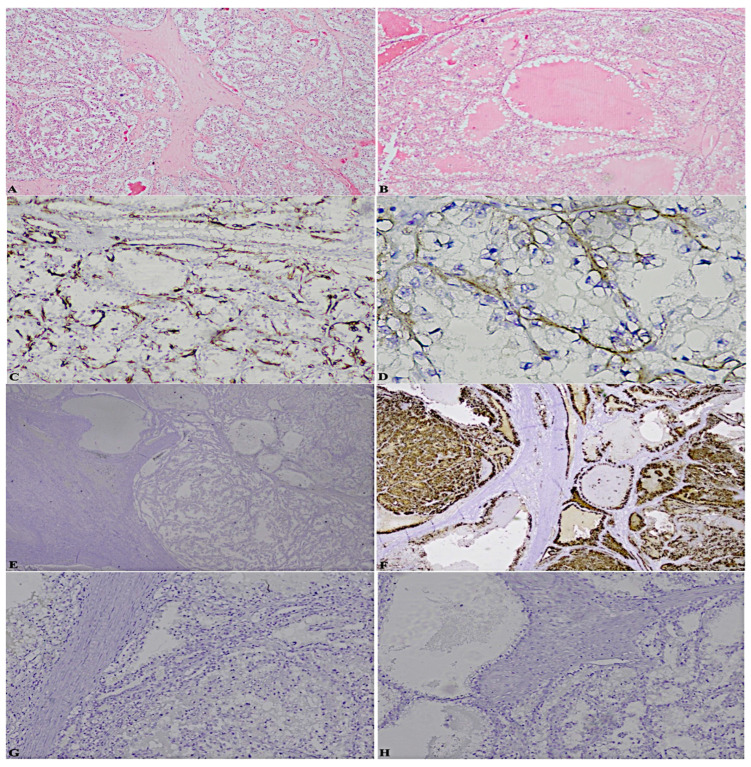
(**A**). Hematoxylin-eosin neoplastic stroma (2×) (**B**). Hematoxylin-eosin thyroid-like areas (2×) (**C**). Patchy CK7+ (10×) (**D**). Mainly cup-shaped CAIX+ (40×) (**E**). CK34betaE12− (2×) (**F**). CD10+++ (2×) (**G**). TTF-1- (2×) (**H**). Thyroglobulin- (2×).

**Figure 3 diagnostics-13-00086-f003:**
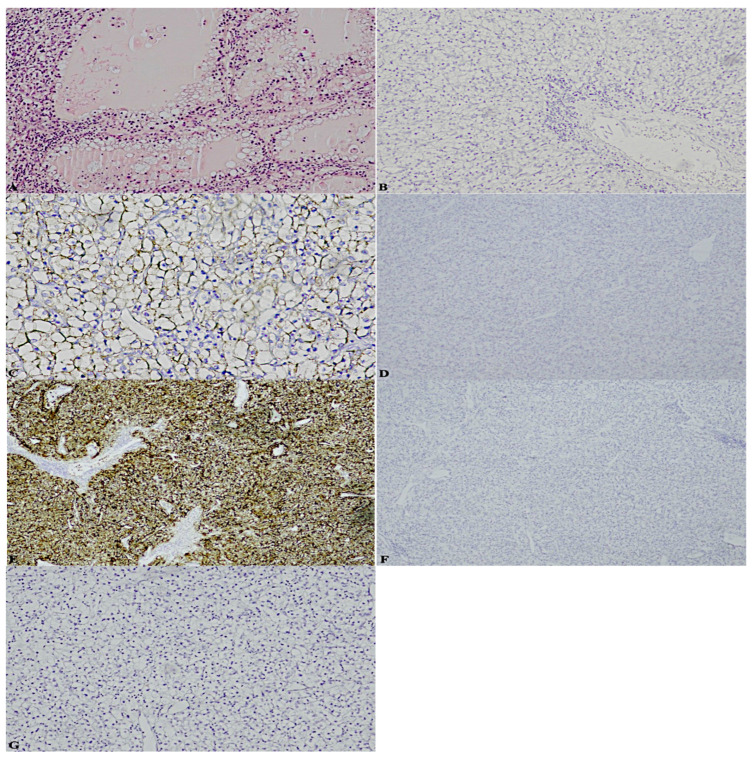
(**A**). Hematoxylin-eosin thyroid-like areas (4×) (**B**). CK7− (10×) (**C**). Box-shaped CAIX+ (20×) (**D**). CK34betaE12− (4×) (**E**). CD10+++ (4×) (**F**). TTF-1- (4×) (**G**). Thyroglobulin- (10×).

**Figure 4 diagnostics-13-00086-f004:**
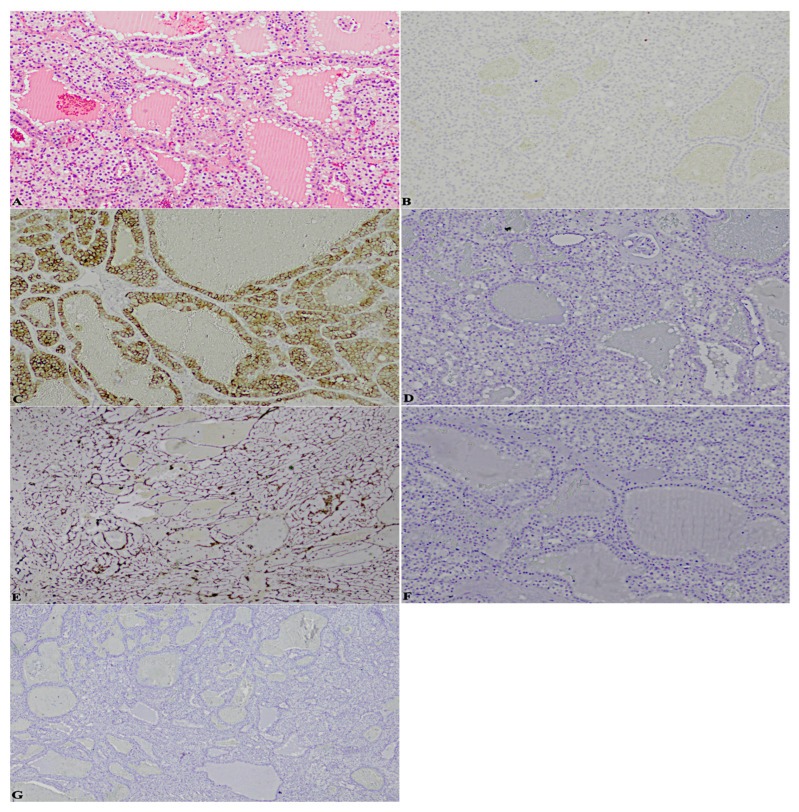
(**A**). Hematoxylin-eosin thyroid-like areas (2×) (**B**). CK7− (2×) (**C**). CD117+ (2×) (**D**). CK34betaE12− (2×) (**E**). Vimentin −/+ (2×) (**F**). TTF-1- (2×) (**G**). Thyroglobulin- (2×).

## Data Availability

Not applicable.

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
