# Peer review of "Are Renal Cell Carcinoma with Fibromyomatous Stroma (RCC-FMS) and Thyroid-like Follicular Carcinoma of the Kidney (TLFCK) Really Independent Variants?"

_diagnostics, 2022, doi:10.3390/diagnostics13010086_

Round 1

Reviewer 1 Report

The authors suggest that RCC-FMS and TLFCK appear to be more appropriate variants of already classified neoplastic entities rather than new independent neoplasms.

The study is so interesting, however, I have some concerns to be discussed.

-Can we make such a classification based on immunohistochemical analysis alone?

-Are you trying to argue that the new classification is erroneous?

Does it have a positive impact on treatment and diagnosis?

-How did you come up with this analysis?

-Is there any change in treatment?

Author Response

Dear Review
Thank you very much for your suggestions that helped us to improve our work

Reviewer 2 Report

The authors aim to better characterize Renal Cell Carcinoma with Fibromyomatosus Stroma and Thyroid-Like Follicular Carcinoma of the Kidney as independent subtypes or morphological variants of well-established renal cell carcinoma subtypes.  

Major comments:

1. In an era of molecular pathology, the study is based primarily on immunohistochemistry. It is already well-documented that there is morphological and immunohistochemical overlap of these subtypes with the commoner RCC variants. As is also the case in other organ systems, molecular events better establish tumor classifications. The lack of molecular analysis in the study should at least be accepted as a major limitation of the study. This is particularly important as some recent studies and review insist on the tumor 'Renal Cell Carcinoma with Fibromyomatosus Stroma' being a distinct entity due to differing molecular features (PMID: 35249990).

2. Results:  The results lack clarity. The way the results are summarized indicates that there were no cases which perfectly matched with the description of RCC-FMS. If that's the case then the study cannot draw a conclusion regarding novelty of this entity. 

3. Results:  Out of 120, there were 11 cases  with TLFCK morphology. The frequency of these tumors in the reported literature and this study needs comparison. Also were these areas focal or diffuse (what was the percentage area with this morphology to the total tumor area sampled)? What was the phenotype of the rest of the tumor? Answers to these queries would have impact on the interpretation of immunohistochemistry results. 

4. Images are not crisp and lack contrast.

5. The status of these tumors in the recent 2022 classification needs to be discussed.

Minor comments:

1. There are multiple grammatical errors. Long sentences are difficult to read. These can be split for shortening.

Author Response

(The authors gave the same response as above.)

Round 2

Reviewer 1 Report

The authors replied well, so the manuscript is suitable for publication.

Author Response

Dear
Thanks again for your comments and suggestions

Reviewer 2 Report

The authors have greatly improved upon the manuscript. However, certain concerns remain:

1. Grammatical errors and 'difficult to read' text are still there.

2. How do the authors explain the discrepancy from the existing literature whereby these entities have been suggested to be distinct entities? Can the conclusions be drawn from their small cohort? Hence, doing molecular analysis would have been preferable at least to document lack of mutations typically associated with clear cell RCC, 

Author Response

(The authors gave the same response as above.)
